# Superb Li-Ion Storage of Sn-Based Anode Assisted by Conductive Hybrid Buffering Matrix

**DOI:** 10.3390/nano13202757

**Published:** 2023-10-13

**Authors:** Jinsil Shin, Sung-Hoon Park, Jaehyun Hur

**Affiliations:** 1Department of Chemical and Biological Engineering, Gachon University, 1342 Seongnam-daero, Seongnam 13120, Republic of Korea; plasticeye76@gmail.com; 2Department of Mechanical Engineering, Soongsil University, 369 Sangdo-ro, Dongjakgu, Seoul 06978, Republic of Korea

**Keywords:** hybrid TiO_2_-C matrix, ternary composite, anode, Li-ion batteries

## Abstract

Although Sn has been intensively studied as one of the most promising anode materials to replace commercialized graphite, its cycling and rate performances are still unsatisfactory owing to the insufficient control of its large volume change during cycling and poor electrochemical kinetics. Herein, we propose a Sn-TiO_2_-C ternary composite as a promising anode material to overcome these limitations. The hybrid TiO_2_-C matrix synthesized via two-step high-energy ball milling effectively regulated the irreversible lithiation/delithiation of the active Sn electrode and facilitated Li-ion diffusion. At the appropriate C concentration, Sn-TiO_2_-C exhibited significantly enhanced cycling performance and rate capability compared with its counterparts (Sn-TiO_2_ and Sn-C). Sn-TiO_2_-C delivers good reversible specific capacities (669 mAh g^−1^ after 100 cycles at 200 mA g^−1^ and 651 mAh g^−1^ after 500 cycles at 500 mA g^−1^) and rate performance (446 mAh g^−1^ at 3000 mA g^−1^). The superiority of Sn-TiO_2_-C over Sn-TiO_2_ and Sn-C was corroborated with electrochemical impedance spectroscopy, which revealed faster Li-ion diffusion kinetics in the presence of the hybrid TiO_2_-C matrix than in the presence of TiO_2_ or C alone. Therefore, Sn-TiO_2_-C is a potential anode for next-generation Li-ion batteries.

## 1. Introduction

The need for high-capacity and high-rate Li-ion batteries (LIBs) has been intensified by the growing demand for various applications that require long-lasting and fast-rechargeable batteries, such as portable electric devices and electric vehicles [1,2,3,4,5,6,7,8,9,10,11,12]. The anode—one of the key components in LIBs—stores or releases energy by hosting/expelling Li ions paired with the cathode. Graphite has been commercialized as a typical anode material owing to its high cycling stability, fast charging, and low self-discharge. However, the low theoretical capacity of graphite (372 mAh g^−1^) is a significant limitation for sustainable use in emerging applications that require high energy density. To resolve this issue, many high-capacity anode materials (e.g., theoretical capacities of 4200 mAh g^−1^ for Si, 2596 mAh g^−1^ for P, 993 mAh g^−1^ for Sn, 1384 mAh g^−1^ for Ge, and 1494 mAh g^−1^ for SnO_2_) have been proposed as alternatives to graphite [13,14,15,16,17].

Sn-based electrodes are excellent candidates for anode materials owing to their high theoretical capacity, low cost, and chemical/mechanical stability. Nevertheless, the practical applications of Sn-based anodes have been hindered by their large volume changes (Li_4.4_ + Sn → Li_4.4_Sn) during lithiation/delithiation, which lead to large capacity losses and degrade the cycling performance [18]. One of the most common strategies for solving this problem is to employ an appropriate secondary component that can buffer the volume expansion of the active material, promoting cycling stability and reversibility. Although this composite approach reduces the overall capacity, the cycling stability can be significantly improved [19,20,21,22,23,24,25,26]. Liu et al. fabricated a spherical-like Sn-Ni alloy composite from SnO_2_, NiO, and Super P carbon via annealing at 900 °C for 2 h, which exhibited a reversible capacity of 448.9 mAh g^−1^ with 78.6% capacity retention after 20 cycles [27]. Hassoun et al. studied SnSb-C nanocomposites prepared using a two-step method (sol-gel and calcination). The electrode exhibited a reversible capacity of 300 mAh g^−1^ at 100 mA g^−1^ after 100 cycles [28]. Youn et al. synthesized N-doped reduced graphene oxide and nanocrystalline tin sulfide composites using a one-step thermal annealing method. The anode exhibited a specific capacity of 562 mAh g^−1^ at 0.2 A g^−1^ after 200 cycles [29].

Despite these efforts, the cycling performance is still not fully satisfactory because of the continuous degradation of the specific capacity, which necessitates the development of better composite materials in which the cycling stability is well-maintained without significant capacity reduction. Recently, the introduction of an inorganic carbon hybrid matrix (e.g., TiC-C) to active materials has been demonstrated to be an effective strategy for restraining the volume expansion of active materials [30,31,32,33,34,35]. This approach exploits the synergistic effect of the binary buffering material, reinforcing the mechanical and electrical properties and outperforming a single-component buffering matrix.

Herein, we propose a ternary composite consisting of Sn-TiO_2_-C as a promising Sn-based anode material for high-performance LIBs. Sn-TiO_2_-C was prepared via two-step high-energy ball milling (HEBM). The cycling instability of active Sn was better regulated by the TiO_2_-C hybrid buffering matrix than by TiO_2_ or C alone. This was because of their synergistic buffering role, which was based on the excellent mechanical properties of TiO_2_ and high electronic conductivity of C. The superiority of Sn-TiO_2_-C to its counterparts (Sn-TiO_2_ and Sn-C) with regard to electrochemical performance and kinetics was experimentally confirmed. In addition, the optimal C content (20 wt%) was experimentally determined for the Sn-TiO_2_-C electrode. Sn-TiO_2_-C exhibited good cycling performance (with a reversible specific capacity of 691 mAh g^−1^ at 200 mA g^−1^ after 100 cycles) and rate performance (446 mAh g^−1^ at 3 A g^−1^). Furthermore, the electrochemical reaction mechanism and kinetics of the Sn-TiO_2_-C composites were studied.

## 2. Materials and Methods

### 2.1. Synthesis of Sn-TiO_2_-C, Sn-TiO_2_, and Sn-C Composite

A ternary composite of Sn-TiO_2_-C was synthesized using a two-step HEBM process. First, SnO_2_ (99.9% trace metal basis, 10 mesh, Sigma-Aldrich, St. Louis, MO, USA), and Ti powder (325 mesh, 99.5% (metal basis), Sigma-Aldrich) were mixed (1:1, mol/mol) and milled at 300 rpm for 30 h in an Ar environment. Then, acetylene black (C) powder (99.9+% S.A. 75 m^2^/g, bulk density of 170–230 g L^−1^, Alfa Aesar, Haverhill, MA, USA) was added to the synthesized material (Sn-TiO_2_) at 300 rpm for 30 h. Three different C contents (10, 20, and 30 wt%) were applied to prepare different Sn-TiO_2_-C (10, 20, and 30 wt%) composite electrodes. They were denoted as Sn-TiO_2_-C (10 wt%), Sn-TiO_2_-C (20 wt%), and Sn-TiO_2_-C (30 wt%), respectively. Binary composites of Sn-TiO_2_ and Sn-C were synthesized as control samples using one-step HEBM. Sn-TiO_2_ was prepared after the first step of HEBM for synthesizing Sn-TiO_2_-C, and Sn-C was prepared by milling a mixture of Sn (99.8%, metal basis, Alfa Aesar) and C (8:2, mol/mol) under the same milling conditions used for Sn-TiO_2_ preparation (denoted as Sn-TiO_2_ (20 wt%)). The mechanochemical synthesis routes for Sn-TiO_2_-C, Sn-TiO_2_, and Sn-C are expressed as follows:(1)$$SnO_{2}+Ti\overset{1st\;HEBM}{\to }\;Sn\left(major\right)+TiO_{2}(SnO_{2};minor)\;\overset{2nd\;HEBM\;(+C)}{\to }\;Sn-TiO_{2}-C\;(10,\;20,\;30\;\mathrm{wt}\%)$$
(2)$$SnO_{2}+Ti\overset{\;HEBM}{\to }\;Sn\left(major\right)-TiO_{2}(SnO_{2};minor)$$
(3)$$Sn+C\overset{\;HEBM}{\to }Sn-C\;(20\;\mathrm{wt}\%)$$

### 2.2. Cell Preparation

A coin-type cell (CR2032, Rotech, Inc., Gwangju, Republic of Korea) was used to evaluate the electrochemical performance of the anode materials. To prepare the binder solution, polyvinylidene fluoride (average MW 534,000 by GPC, powder, Sigma-Aldrich) was dissolved in anhydrous N-methylpyrrolidone (99.5%, Sigma-Aldrich) for 24 h (12 wt%). A slurry was prepared by dispersing the active material and conductive C (Super P, 99+%, metal basis, Alfa Aesar) in the binder-dissolved solution such that the weight ratio of the active material, binder, and conductive C was 70:15:15, respectively, followed by stirring at 150 rpm for 24 h. The prepared slurry was casted on Cu foil, followed by drying in a vacuum oven at 70 °C for 24 h. The dried working electrode was punched into a circular shape (diameter: 1.25 cm). A typical loading amount of the electrode material was 1.5 mg. Polyethylene and Li foil (99.9%, metal basis, 0.75 mm thickness, Alfa Aesar) were used as the separator and counter electrode, respectively. Additionally, 1.5 mL of the electrolyte (1 M LiPF_6_ dissolved in ethylene carbonate/diethylene carbonate, 1:1, *v*/*v*) was added during cell assembly. The cells were assembled in an Ar-gas-filled glovebox.

### 2.3. Characterization

X-ray diffraction (XRD; Rigaku D/MAX-2200, Rigaku Corporation, Akishima, Japan) equipped with Cu-Kα radiation was used to examine the crystalline structures of the samples. High-resolution transmission electron microscopy (HRTEM; JEOL JEM-2100F, JEOL Ltd., Tokyo, Japan) was used to identify the synthesized components of the composite. The morphology of the composite was examined using scanning electron microscopy (SEM; Hitachi SU8600, Hitachi High-Tech Corporation, Toranomon, Japan) at the Smart Materials Research Center for IoT at Gachon University. The elemental distribution was determined via energy-dispersive X-ray spectroscopy (EDS) mapping. X-ray photoelectron spectroscopy (XPS; Thermo Fisher Scientific Alpha XPS system, Thermo Fisher Scientific, Cambridge, UK) was used to analyze the binding energies of the elements. Calibration was conducted using the C 1s method, in which the binding energy of C 1s was calibrated at 284.6 eV. Brunauer–Emmett–Teller (BET: Micromeritics (ASAP 2020), Micromeritics Instrument Corporation, Norcross, GA, USA) analysis was performed to identify the surface area and porosity of the samples.

### 2.4. Electrochemical Measurement

The voltage profile, rate capability, and cycling performance were measured using a battery cycler (WBCS 3000, WonAtech, Seoul, Republic of Korea). Cyclic voltammetry (CV; WonAtech ZIVE MP1, WonAtech, Seoul, Republic of Korea) and ex situ XRD (Rigaku D/MAX-2200, Japan) were used to analyze the electrochemical reactions during charging and discharging. The Li-ion diffusion kinetics were analyzed using electrochemical impedance spectroscopy (EIS; ZIVE MP1, Republic of Korea) in the frequency range of 1–1000 kHz.

## 3. Results and Discussion

### 3.1. Characterization of As-Synthesized Sn-TiO_2_-C

Figure 1a shows the XRD patterns of Sn-TiO_2_-C (20 wt%) and Sn-TiO_2_. The main 2θ peaks observed at 30.6°, 32.0°, 43.9°, and 44.9° for Sn-TiO_2_ corresponded to the (2 0 0), (1 0 1), (2 2 0), and (2 1 1) planes, respectively, of Sn (PDF# 04-0673). The minor peaks observed at 26.6°, 35.9°, 41.2°, and 54.3° (magnified view in Appendix A) were assigned to TiO_2_ (rutile, PDF# 21-1276). From these results, it is evident that the O atoms were successfully transferred from Sn to Ti ($SnO_{2}+Ti\overset{\;HEBM}{\to }\;Sn-TiO_{2}$) via a mechanochemical reaction, during which Sn and TiO_2_ nanoparticles (NPs) were homogeneously mixed with a reduced size. Minor peaks corresponding to SnO_2_ (cassiterite) were detected, suggesting that some of the SnO_2_ (cassiterite) NPs were not completely converted into Sn and TiO_2_. This may have been associated with the lower standard enthalpy of SnO_2_ compared with that of TiO_2_ [30,31]. Even when the milling time was increased to 30 h, the SnO_2_ was not completely converted into Sn (Appendix A). Nevertheless, the coexistence of Sn and SnO_2_ in the Sn-SnO_2_ active material is not considered detrimental, because the theoretical capacity of SnO_2_ is even higher than that of Sn. In addition, Sn and SnO_2_ were expected to mix well during HEBM, with a significantly increased surface area relative to the pristine powder state. For Sn-TiO_2_-C (20 wt%), although the peaks were broadened relative to Sn-TiO_2_ owing to the presence of amorphous C, all the peak positions were identical, indicating that Sn and TiO_2_ were only physically mixed with C, without any changes in their chemical properties during the second milling process.

The presence and distributions of Sn, TiO_2_, and C in the synthesized Sn-TiO_2_-C (20 wt%) were characterized using HRTEM, SEM, and EDS. As shown in Figure 1b, the presence of Sn (red dotted region, with an interplanar distance of 0.292 nm from the (2 0 0) plane) and TiO_2_ (black dotted region, with an interplanar distance of 0.325 nm from the (1 1 0) plane) surrounded by amorphous C (orange dotted region) was confirmed in the HRTEM image. Unreacted SnO_2_ was also detected, for which the crystallite size was significantly reduced (~5−10 nm) (Appendix A) compared with its pristine powder size (~1900 μm according to the vendor). During HEBM, although some SnO_2_ particles were not completely converted to Sn and TiO_2_, their sizes were significantly reduced, and they were better mixed with the produced Sn and TiO_2_, which was beneficial for obtaining a high capacity and stable electrochemical reaction. The average particle size of the as-synthesized Sn-TiO_2_-C (20 wt%) was ~305 nm (Appendix A), which was far smaller than the sizes of the precursor materials (SnO_2_: 1900 μm, Ti: 44 μm) after HEBM. The uniform distributions of constituent atoms (Sn, Ti, C, and O) in the product was confirmed with EDX mapping (Figure 1c–g). BET analysis was performed to characterize the porosity and surface area of Sn-TiO_2_-C (20 wt%), Sn-TiO_2_, and Sn-C (20 wt%). Appendix A show the adsorption and desorption of the N_2_ curves of Sn-TiO_2_-C (20 wt%), Sn-TiO_2_, and Sn-C (20 wt%). The measured BET surface areas of Sn-TiO_2_-C (20 wt%), Sn-TiO_2_, and Sn-C (20 wt%) were 45.386, 3.3553, and 50.5148 m^2^ g^−1^, respectively (Appendix A). The higher surface areas of Sn-TiO_2_-C (20 wt%) and Sn-C (20 wt%) than those of Sn-TiO_2_ were mainly associated with the micropores existing in the C. The larger surface area increased the contact area between the active material and the electrolyte, allowing facile Li-ion exchange and effective buffering during cycling. The pore distribution is depicted in S5d-S5f. In the case of Sn-TiO_2_-C (20 wt%), both the micropores and mesopores were almost equally distributed. However, only the micropores were overwhelmingly observed for Sn-C (20 wt%). For Sn-TiO_2_, although it had both micropores and mesopores, their surface areas were relatively smaller than that of Sn-TiO_2_-C (20 wt%). The presence of micropores and mesopores at an appropriate ratio in Sn-TiO_2_-C (20 wt%) was expected to be favorable for effective Li-ion diffusion and increase the reaction sites while accommodating the volume change of active material.

Figure 2 shows the XPS spectra of the as-synthesized Sn-TiO_2_-C (20 wt%). The survey spectrum confirmed the presence of Sn, Ti, O, and C in the sample (Figure 2a). In the high-resolution Sn 3d spectrum, the fitted peaks at 487.2 and 495.6 eV corresponded to Sn 3d_3/2_ and Sn 3d_5/2_, respectively. The peak splitting of 8.4 eV in the Sn 3d spectrum indicated the presence of Sn (Figure 2b), which was consistent with the XRD results (Figure 1a) [36]. The peaks at 464.9 and 459.2 eV with orbital splitting of 5.7 eV in the Ti 2p spectrum corresponded to Ti 2p_1/2_ and Ti 2p_3/2_, respectively, and the presence of TiO_2_ in Sn-TiO_2_-C (20 wt%) (Figure 2c). In the O 1s spectrum, the deconvoluted peaks at 530.4, 531, 532.1, and 533.3 eV corresponded to the Ti-O, Sn-O, C-O, and O-H functional groups, respectively (Figure 2d). The deconvoluted peaks at 284.6, 286.3, and 289.3 eV in the C 1s spectrum corresponded to C-C, C-O-C and C-O=C, respectively, in Sn-TiO_2_-C (20 wt%) (Figure 2e). Comparing the XPS spectra of Sn-TiO_2_-C (20 wt%) with those of Sn-TiO_2_ (Appendix A) revealed that most peaks were identical, except for the C 1s spectrum, where the intensities of the C-O-C and C-O=C peaks were significantly reduced owing to the absence of TiO_2_-C in Sn-TiO_2_.

### 3.2. Electrochemical Reaction Mechanism and Kinetics of Sn-TiO_2_-C Anode

The electrochemical reaction mechanism of Sn-TiO_2_-C (20 wt%) was studied using CV, voltage profiles, and ex situ XRD. Figure 3a presents the CV profile of Sn-TiO_2_-C (20 wt%) obtained at a voltage scan rate of 0.2 mV s^−1^ for the first five cycles. During the first discharging cycle, three typical lithiation processes of the main active material Sn proceeded at ~0.89 V (2Li^+^ + 2e^−^ + 5Sn → Li_2_Sn_5_), ~0.63 V (31Li^+^ + 2Li_2_Sn_5_ + 31e^−^ → 5Li_7_Sn_2_), and ~0.27 V (9Li^+^ + 5Li_7_Sn_2_ + 9e^−^ → 2Li_22_Sn_5_). The peak observed at ~1.2 V was attributed to the partial electrolyte decomposition catalyzed by Sn [12,37,38]. A broad peak close to ~0 V indicated the lithiation of C and the formation of a solid–electrolyte interphase (SEI). During the first charge cycle, a series of intermediate products were observed: Li_22_Sn_5_, Li_7_Sn_3_, LiSn, and Li_2_Sn_5_. The oxidation peak at ~1.2 V was related to the delithiation of Li-C [39]. These electrochemical reactions well-matched the capacity plateaus in the voltage profile shown in Figure 3b. In the ex situ XRD analysis (Figure 3c), only Li_22_Sn_5_ was detected as the main lithiation product in the fully discharged state. Notably, no other lithiation products were observed, probably because of their small amounts and surface amorphization during lithiation [40]. In the fully charged state, the active Sn peaks clearly reemerged, indicating reversible crystalline evolution of the active material during cycling. After the first cycle, the discharge and charge peaks were broader and smaller, but the peak intensities and positions remained almost unchanged, indicating reversible reactions of Sn-TiO_2_-C (20 wt%).

Considering these results, the electrochemical reaction mechanism of Sn-TiO_2_-C (20 wt%) can be summarized as follows:

Lithiation
(4)$$5Sn+2{Li}^{+}+2e^{-}\to \;{Li}_{2}{Sn}_{5}\;(\sim 0.89\;V)$$
(5)$${2Li}_{2}{Sn}_{5}+31{Li}^{+}+31e^{-}\;\to \;{5Li}_{7}{Sn}_{2}\;(\sim 0.63V)$$
(6)$${{5Li}_{7}{Sn}_{2}+9{Li}^{+}+9e^{-}\to 2Li}_{22}{Sn}_{5}\;(\sim 0.27V)$$

Delithiation
(7)$${3Li}_{22}{Sn}_{5}\to {5Li}_{7}{Sn}_{3}+{31Li}^{+}+31e^{-}\;(\sim 0.66V)$$
(8)$${Li}_{7}{Sn}_{3}\to 3LiSn+{4Li}^{+}+4e^{-}\;(\sim 0.75\;V)$$
(9)$$5LiSn\to {Li}_{2}{Sn}_{5}+3{Li}^{+}+3e^{-}\;(\sim 0.81\;V)$$
(10)$${Li}_{2}{Sn}_{5}\to 5Sn+{2Li}^{+}+2e^{-}\;(\sim 3V)$$

Figure 4a shows the CV profiles of Sn-TiO_2_-C (20 wt%) at various scan rates (0.2–1.4 mV s^−1^). To identify the contribution of capacitive and diffusion-controlled reactions, three distinctive peaks in the delithiation process were selected (peak 1: ~0.60 V, peak 2: ~0.7 V, and peak 3: ~1.3 V; peaks 1 and 2 corresponded to the delithiation of Li_y_Sn and peak 3 corresponded to the delithiation of Li_x_C) [41]. From the log *i* (current) vs. log *ν* (scan rate) plot (Figure 4b), the slope b was obtained via linear fitting of the following equation [42]:(11)log⁡i=blog⁡υ+log⁡a 
where the *b* values of 1 and 0.5 reflect the surface capacitive and diffusion-controlled systems, respectively. The closer the value of b is to 0.5, the more it can be interpreted as diffusion control, and the closer it is to 1, the more capacitive-dominant it is. The *b* values for peaks 1, 2, and 3 are 0.60, 0.61, and 0.67, respectively. These results indicated that the reactions for peaks 1, 2, and 3 are all mixed reactions of surface and diffusion control. The quantitative contributions of the capacitive and diffusion-controlled reactions can be calculated by linearly fitting the *i v^−^*^1/2^ vs. *v*^1/2^ plot using the following equation [43,44]:(12)$$i\;{\;\upsilon }^{-1/2}=k_{1}\;\upsilon^{1/2}+k_{2}$$
where *k*_1_ and *k*_2_ are the parameters corresponding to the capacitive and diffusion-controlled behaviors, respectively. As shown in Figure 4c–e, the contribution from capacitive storage (red bars in the graphs) consistently increased with an increase in the scan rate, regardless of the peak type. This was because the number of Li ions that could diffuse into the internal electrode was reduced at high-voltage scan rates, reducing the overall capacity. However, the presence of a capacitive reaction allowed fast Li-ion charge-transfer kinetics, enhancing the rate performance. For peaks 1–3, the contributions from the capacitive and diffusion-controlled reactions were 19.22%/80.78%, 19.80%/80.20%, and 22.33%/77.67%, respectively, at a scan rate of 0.2 mV s^−1^. The appropriate capacitive and diffusion-controlled behaviors of Sn-TiO_2_-C (20 wt%) were expected to balance the electrochemical kinetics and overall capacity.

### 3.3. Electrochemical Performance of Sn-TiO_2_-C Anode

Figure 5a presents the cycling performance of Sn-TiO_2_-C (20 wt%) in comparison with that of its counterparts (Sn-TiO_2_ and Sn-C (20 wt%)) measured at a current density of 200 mA g^−1^. In the first cycle, Sn-C (20 wt%) exhibited a higher specific capacity (998 mAh g^−1^) than Sn-TiO_2_-C (20 wt%) (990 mAh g^−1^) and Sn-TiO_2_ (628 mAh g^−1^). This was reasonable considering the theoretical capacities of the samples (Sn-TiO_2_-C (20 wt%):657 mAh g^−1^, Sn-TiO_2_:728 mAh g^−1^, and Sn-C (20 wt%):869 mAh g^−1^) (Appendix A). The specific capacities of the samples exceeded their theoretical capacities in the first cycle owing to the additional capacity contribution from the SEI layer. As the number of cycles increased, the capacity variation tended to significantly change among the samples. The capacity decreased with an increasing number of cycles for Sn-C (20 wt%), whereas it was stably maintained for Sn-TiO_2_-C (20 wt%), except for some capacity drop during the initial several cycles. After 100 cycles, the specific reversible discharge capacity of Sn-TiO_2_-C (20 wt%) was 669 mAh g^−1^, which was significantly higher than that of Sn-C (20 wt%) (247 mAh g^−1^). For Sn-TiO_2_, the capacity rapidly decreased in the first 10 cycles because of the absence of a buffering C matrix. Rate capability measurements revealed a similar trend (Figure 5b). Sn-C (20 wt%) exhibited a higher capacity than Sn-TiO_2_-C (20 wt%) and Sn-TiO_2_ at 0.1 A g^−1^, as it had the highest theoretical capacity. However, when a higher current density was applied, Sn-TiO_2_-C (20 wt%) exhibited superior performance to the other samples, owing to its efficient and stable Li-ion transport. The average discharge capacities of Sn-TiO_2_-C (20 wt%) were 586, 563, 540, and 439 mAh g^−1^ at 0.1, 0.5, 1.0, and 3.0 A g^−1^, respectively. After the original current density recovered to 0.1 A g^−1^, the capacity retention of Sn-TiO_2_-C (20 wt%) was as high as 96%, indicating its reversible and resilient rate performance. Figure 5c shows the cycling performance of Sn-TiO_2_-C with respect to the C content (10, 20, and 30 wt%). Although Sn-TiO_2_-C (20 wt%) and Sn-TiO_2_-C (30 wt%) exhibited stable cycling behavior after several initial cycles, Sn-TiO_2_-C (10 wt%) exhibited continuous capacity fading owing to the insufficient buffering effect of 10 wt% C in the composite. Sn-TiO_2_-C (20 wt%) outperformed Sn-TiO_2_-C (30 wt%) because of its higher active Sn content. The reversible specific discharge capacity of Sn-TiO_2_-C (20 wt%) was 669 mAh g^−1^ after 100 cycles at 200 mA g^−1^, which was higher than that of Sn-TiO_2_-C (30 wt%) (543 mAh g^−1^), suggesting that the optimal C content of the Sn-TiO_2_-C composite was 20 wt%. Overall, Sn-TiO_2_-C (20 wt%) outperformed the most recently studied Sn-based anodes for LIBs (Appendix A). Figure 5d–f present the voltage profiles of Sn-C, Sn-TiO_2_, and Sn-TiO_2_-C (20 wt%) from the 1st to the 100th cycle. The initial coulombic efficiencies (CEs) of Sn-C (20 wt%), Sn-TiO_2_, and Sn-TiO_2_-C (20 wt%) were 73.5%, 63.5%, and 81.9%, respectively (Appendix A). As the cycle number increased, the increase in CE was more pronounced for Sn-TiO_2_-C (20 wt%) than for the other samples. For example, the CE of Sn-TiO_2_-C (20 wt%) after 10 cycles was 98.3%, which was higher than those of Sn-C (20 wt%) (94.4%) and Sn-TiO_2_ (96.2%). This indicated the highly reversible lithiation/delithiation of Sn-TiO_2_-C (20 wt%), which stabilized the cycling performance with an increase in the number of cycles, as shown in Figure 5a,c. To gain further insight into the good performance of Sn-TiO_2_-C (20 wt%), EIS was performed. Figure 5g shows the Nyquist plots and equivalent circuits (R_ct_: charge-transfer resistance; R_s_: electrolyte resistance; CPE: constant phase element related to the double-layer capacitance; and *Z_w_*: Warburg impedance) of Sn-C (20 wt%), Sn-TiO_2_, and Sn-TiO_2_-C (20 wt%) in the frequency range of 1–1000 kHz after 20 cycles. From the fitting results (Appendix A), Sn-TiO_2_-C (20 wt%) had the smallest R_ct_ value, implying it had the most efficient charge transport. Figure 5h shows the plot of real impedance ($Z^{\prime }$) vs. $\omega^{-1/2}$ obtained from EIS measurements for the low frequency range (10 Hz–1 Hz). The Warburg parameter (*σ*) was extracted via linear fitting with the following equation:(13)$$Z^{\prime }=R_{s}+R_{ct}+\sigma \omega^{-1/2}.$$

As the result of linear fitting, the σ values of Sn-TiO_2_-C (20 wt%), Sn-TiO_2_, and Sn-C (20 wt%) were 254.09, 420.51, and 284,47, respectively (Figure 5h). The Li-ion diffusion coefficient (*D_Li_*_+_) can be obtained from the estimated *σ* via the following equation [45]:(14)$$D_{{Li}^{+}}=\frac{R^{2}T^{2}}{2A^{2}n^{4}F^{4}C^{2}\sigma^{2}}$$
where *R*, *T*, *A*, *n*, *F*, and *C* represent the gas constant (8.314 J K^−1^ mol^−1^), absolute temperature (298.15 K), electrode area (1.23 cm^2^), number of electrons per molecule (*n* = 1 mol), Faraday constant (96,485 C mol^−1^), and Li-ion concentration (1 mol cm^−3^), respectively. According to the results, the D_Li+_ of Sn-TiO_2_-C (20 wt%) was higher (3.6 × 10^−13^ cm^2^ s^−1^) than those of Sn-C (20 wt%) (2.9 × 10^−13^ cm^2^ s^−1^) and Sn-TiO_2_ (1.3 × 10^−13^ cm^2^ s^−1^) after 20 cycles, indicating that Sn-TiO_2_-C (20 wt%) had a superior Li^+^ ion diffusion capability compared with the other samples (Figure 5i).

Figure 6a shows the long-term cycling performance of the Sn-based composite electrodes at 500 mA g^−1^. Similar to the cycling performance measured at a lower current density (200 mA g^−1^), the Sn-TiO_2_-C (20 wt%) electrode exhibited significantly better performance than its counterparts at a high current density (500 mA g^−1^) during 500 cycles. The reversible specific capacity of Sn-TiO_2_-C (20 wt%) was 651 mAh g^−1^ after 500 cycles. For Sn-C (20 wt%), the capacity fading was more severe at this high current density than at a low current density because of the insufficient buffering role of the single C matrix. Notably, the capacity of Sn-TiO_2_-C (20 wt%) gradually increased after approximately 100 cycles, which was likely associated with the activation of the electrode and electrolyte decomposition, where more active sites in the electrode were exposed or occupied by Li ions [46]. Figure 6b–g present SEM images of Sn-C (20 wt%), Sn-TiO_2_-C (20 wt%), and Sn-TiO_2_ in the pristine state and after 100 cycles. Although several cracks and large agglomerated particles were observed for Sn-C (20 wt%) and Sn-TiO_2_, the morphological change in Sn-TiO_2_-C (20 wt%) after cycling was relatively insignificant, indicating the mechanical robustness of Sn-TiO_2_-C (20 wt%) supported by the hybrid matrix.

## 4. Conclusions

We developed a ternary Sn-TiO_2_-C composite as a potential LIB anode material. Despite many previous studies on Sn as an active material for LIBs, our work suggested the following new aspects: (i) the homogeneous distribution of its constituents (Sn (SnO_2_ as a minor component), TiO_2_, and C) without much aggregation was confirmed by various analyses (XRD, XPS, SEM, TEM, and EDS); (ii) the synergistic effects of TiO_2_ and amorphous C on active Sn-based electrode performance were comprehensively investigated using various analyses (CV, EIS, and ex situ SEM) where they effectively mitigated the volume change of active Sn during cycling and enhanced the reversible electrochemical reactions; (iii) the enhanced kinetic properties of Sn-TiO_2_-C compared with its counterparts (Sn-C and Sn-TiO_2_) were elucidated by EIS and CV measurement; (iv) the optimum C content in Sn-TiO_2_-C that can maximize the reversible capacity while retaining the cycling stability was found. Consequently, Sn-TiO_2_-C (20 wt%) delivered 669 mAh g^−1^ (after 100 cycles at 200 mA g^−1^) and 651 mAh g^−1^ (after 500 cycles at 500 mA g^−1^), with 99% capacity retention at 200 mA g^−1^ relative to 500 mA g^−1^. The superiority of Sn-TiO_2_-C to Sn-TiO_2_ and Sn-C was associated with enhanced Li-ion diffusion properties, as verified with EIS. This study provides valuable insights for the development of high-performance anode materials for LIBs.

## Figures and Tables

**Figure 1 nanomaterials-13-02757-f001:**
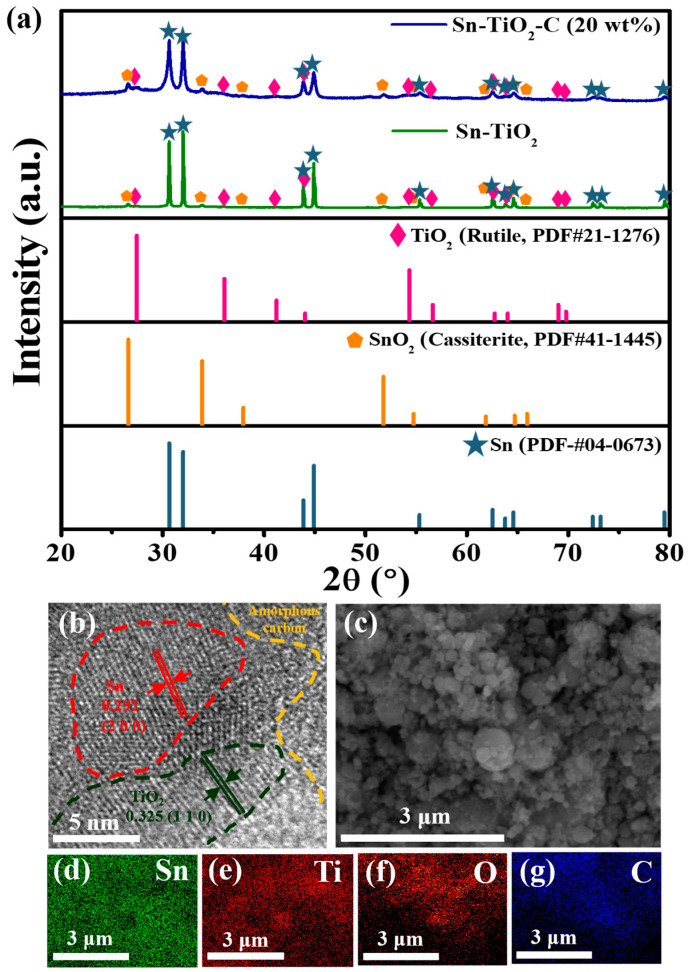
(**a**) XRD spectra of Sn-TiO_2_-C (20 wt%) and Sn-TiO_2_ along with theoretical peaks of TiO_2_ (rutile, PDF# 21-1276), SnO_2_ (cassiterite, PDF# 41-1445), and Sn (PDF# 04-0673). (**b**) HRTEM and (**c**) SEM images of Sn-TiO_2_-C (20 wt%). (**d**–**g**) EDS element maps of Sn-TiO_2_-C (20 wt%) for Sn, Ti, C, and O.

**Figure 2 nanomaterials-13-02757-f002:**
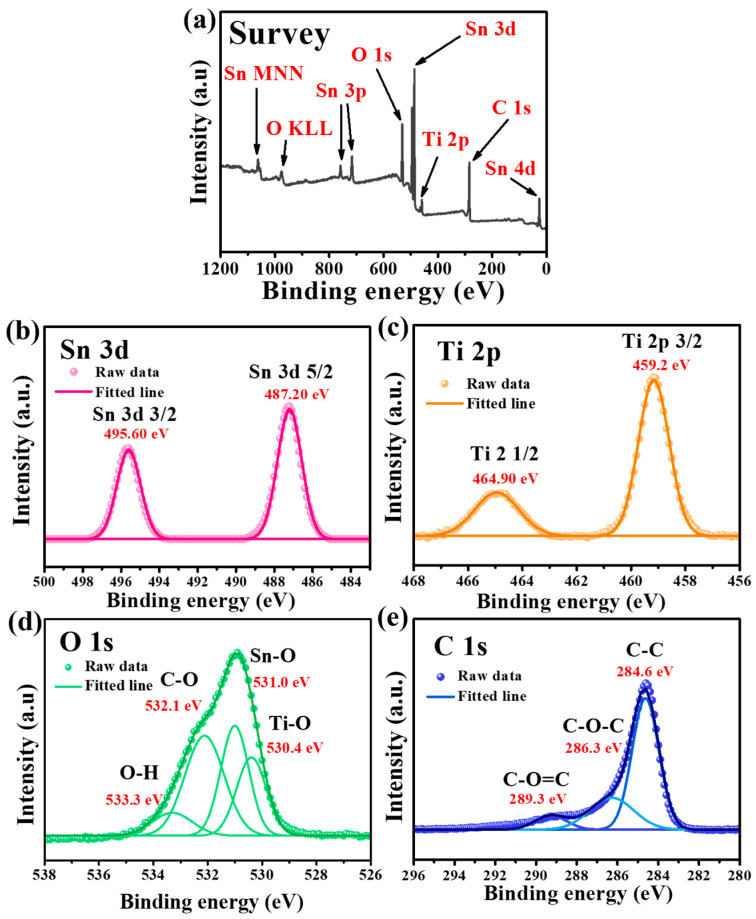
XPS spectra of Sn-TiO_2_-C (20 wt%) powder: (**a**) survey and high-resolution spectra for (**b**) Sn, (**c**) Ti, (**d**) O, and (**e**) C.

**Figure 3 nanomaterials-13-02757-f003:**
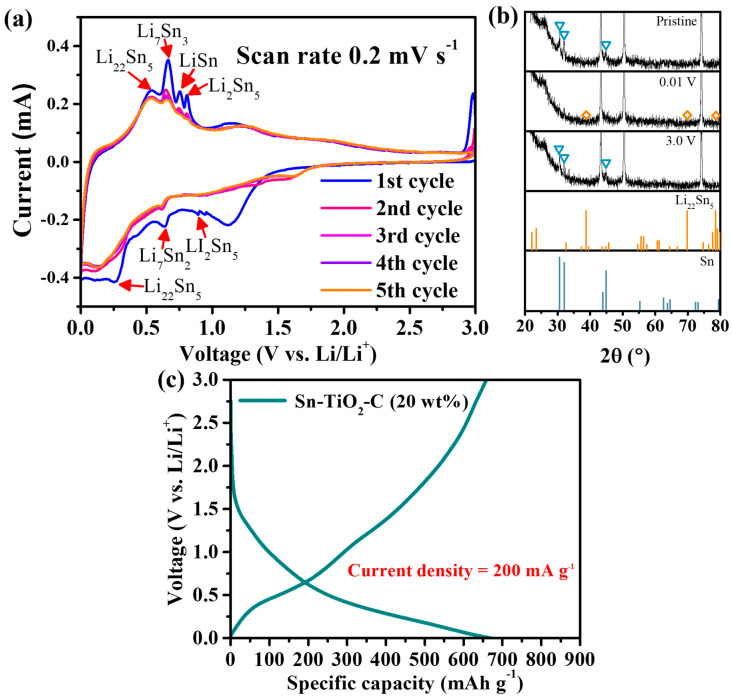
(**a**) CV results; (**b**) ex situ XRD (before cycling and at 0.01 and 3 V) results (triangles and diamond symbols correspond to the peaks from Sn and Li_22_Sn_5_, respectively); (**c**) voltage profile of Sn-TiO_2_-C (20 wt%) after 100th cycle at 200 mA g^−1^.

**Figure 4 nanomaterials-13-02757-f004:**
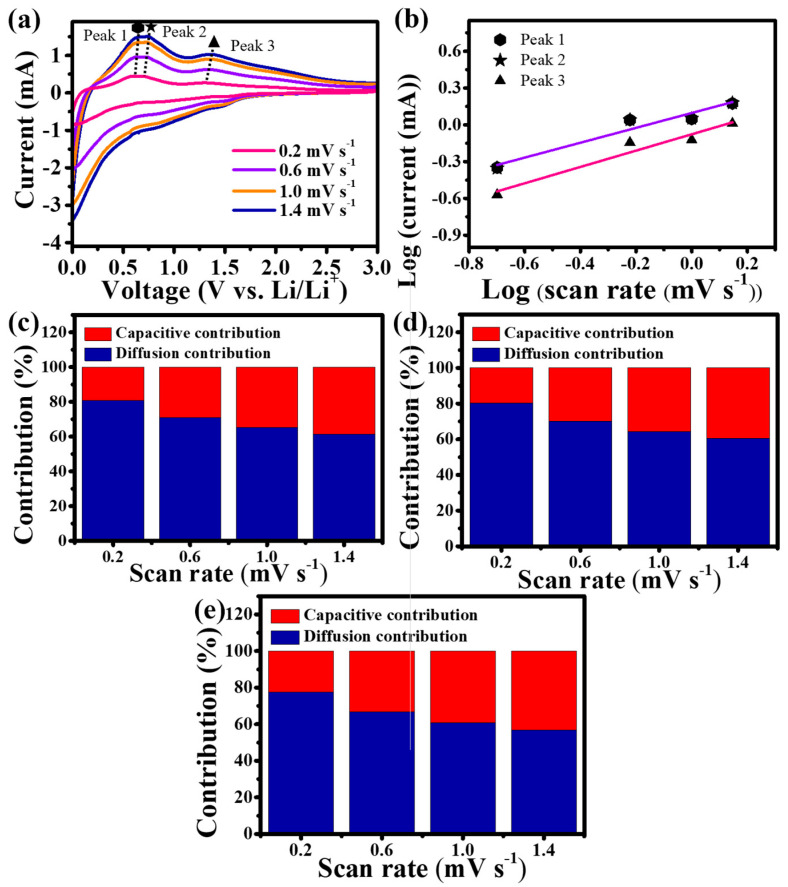
(**a**) CV of Sn-TiO_2_-C (20 wt%) at various scan rates; (**b**) fitted plot of the current variation with respect to the scan rate; proportions of capacitive and diffusion storage contributions at (**c**) peak 1, (**d**) peak 2, and (**e**) peak 3.

**Figure 5 nanomaterials-13-02757-f005:**
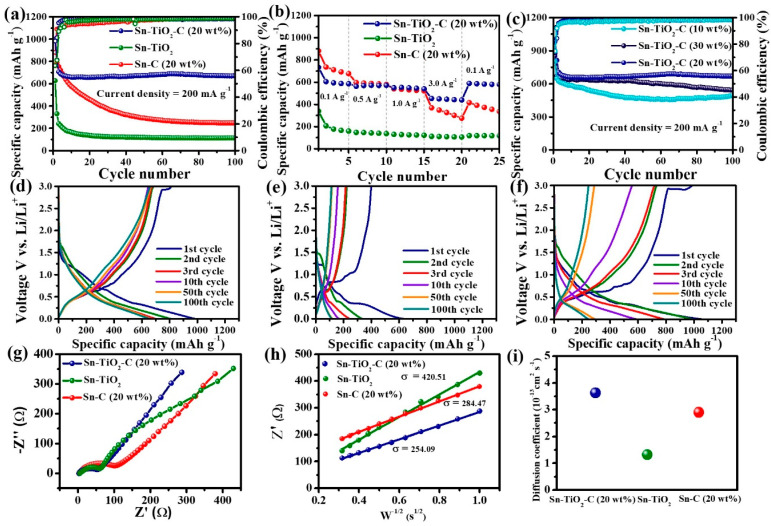
(**a**) Cycle performance (at 200 mA g^−1^); (**b**) rate performance (0.1−3.0 A g^−1^) of Sn-C (20 wt%), Sn-TiO_2_, and Sn-TiO_2_-C (20 wt%); (**c**) cycle performance of Sn-TiO_2_-C (10 wt%), Sn-TiO_2_-C (20 wt%), and Sn-TiO_2_-C (30 wt%) at 200 mA g^−1^; voltage profiles of (**d**) Sn-C (20 wt%), (**e**) Sn-TiO_2_, and (**f**) Sn-TiO_2_-C (20 wt%) at 200 mA g^−1^; (**g**) Nyquist plot and (**h**) fitting plot of Z’ vs. W^−1/2^; (**i**) diffusion coefficients of Sn-C (20 wt%), Sn-TiO_2_, and Sn-TiO_2_-C (20 wt%).

**Figure 6 nanomaterials-13-02757-f006:**
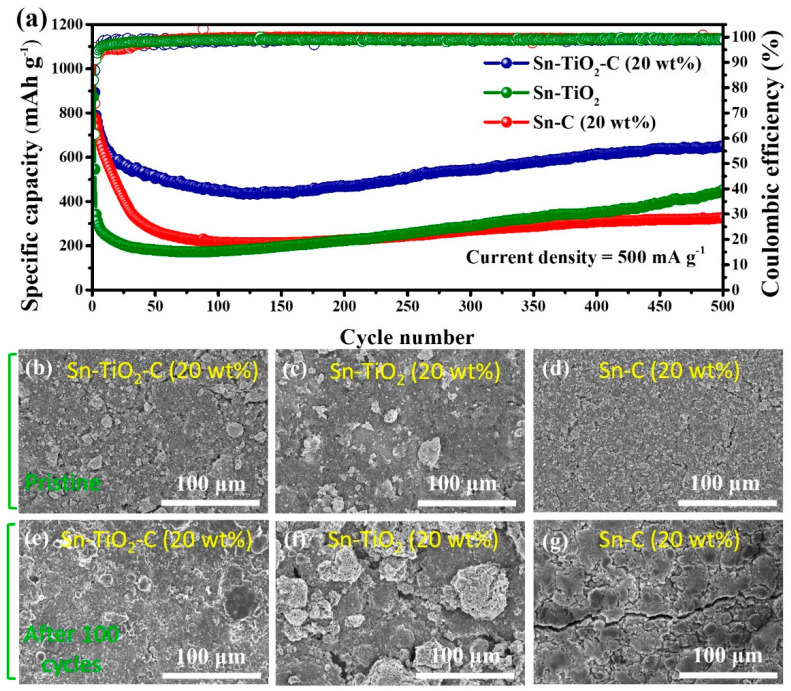
(**a**) Cycle performance of Sn-TiO_2_-C (20 wt%), Sn-TiO_2_, and Sn-C (20 wt%) at 500 mA g^−1^. SEM images of (**b**,**e**) Sn-TiO_2_-C (20 wt%), (**c**,**f**) Sn-TiO_2_, and (**d**,**g**) Sn-C (20 wt%); (**b**–**d**) pristine and (**e**–**g**) after 100 cycles (applied current density of 500 mA g^−1^).

## Data Availability

The data are available on reasonable request from the corresponding author.

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
