# Peer review of "Superb Li-Ion Storage of Sn-Based Anode Assisted by Conductive Hybrid Buffering Matrix"

_nanomaterials, 2023, doi:10.3390/nano13202757_

Round 1

Reviewer 1 Report

Shin et al studied Li-ion storage properties of Sn-based anode by conductive hybrid buffering matrix. The work shows interesting results and can be considered for acceptance after proper revisions.

1. More TEM/HRTEM images should be provided in the main text or supporting files to make related conclusion more solid.

2. Fig.1a. Please use different marks in the xrd pattern to distinguish the different phases, rather than just giving the crystal face index.

3. Fig.1b. Scale bar should be added.

4. Fig.2a. What elements do the other peaks in the full spectrum represent? Please label them.

5. BET tests should be carried out.

6. Introduction. More Sn related materials for LIBs should be described and referenced. The following literature is for your reference.

[1] Rare Metals 2022,41,1504–1511. http://doi.org/10.1007/s12598-021-01886-y  

[2] Journal of Colloid and Interface Science 2021,602,563–572. https://doi.org/10.1016/j.jcis.2021.06.046

[3] Battery Energy 2023,2,20220032. https://doi.org/10.1002/bte2.20220032

7. The format of the paper should meet the requirements of the journal.

Minor editing of English language required.

Reviewer 2 Report

In this work, authors prepared Sn-TiO2-C ternary composite via a common ball milling method. Although the materials and their preparation methods are not new, the materials exhibit satisfactory Li storage performance. Authors also performed comprehensive characterization methods and Li storage tests to the materials. The improved cycling performance of Sn was also detailed confirmation. The reviewer suggests the acceptance of this paper after addressing the following several questions:

1, author should give the reasons why the capacity of composite materials decreases first but then increases.

2, the scale bar of the HRTEM should be added.

3, in Fig. 3c, the crystalline phase evolution of Sn via the variation of voltage is generally accepted, and therefore it should be deleted.

4, some related literatures (Adv. Energy Mater. 2023, 2300453; Appl. Phys. Lett. 2022, 121, 153902) should be cited.

The quality of the English language is acceptable.

Reviewer 3 Report

In the literature we find a nanostructured Sn/TiO2/C composite was prepared from SnO, Ti, and carbon powders using a mechanochemical reduction method and evaluated as an anode material in rechargeable Li-ion batteries. The Sn/TiO2/C nanocomposite was composed of uniformly dispersed nanocrystalline Sn and rutile TiO2 in amorphous carbon matrix. In addition, electrochemical Li insertion/extraction in rutile TiO2 was examined by ex situ XRD and extended X-ray absorption fine structure. The Sn/TiO2/C nanocomposite exhibited excellent electrochemical performance, which highlights its potential as a new alternative anode material in Li-ion batteries.

My questions and comments:

1)What is new in this work, since such a system has already been analyzed?

2)the system we are analyzing is unclear - i.e. percentage differences in the system, there is no detailed description of the samples and its comparison with the symbolism that is in the description of the drawings

3)are reactions 1-3 correct?

4)I think that Figure 3c is bad, it does not show the relationship between capacity and potential. The transformations illustrated during charging and discharging require explanation

5)the description of figure 4 is unclear

6)figure 5i - I don't understand the three measurement points

7)the conclusions should be edited because they are not very precise

8)Literature should be completed

After completing the comments, the work may be published

Round 2

Reviewer 1 Report

The authors addressed all of the reviewers' concerns and made changes as requested. In this case, I suggest that this article be considered for acceptance. 

Reviewer 2 Report

After revision, the manuscript is recommend to publication